# *Corynebacterium glutamicum* Regulation beyond Transcription: Organizing Principles and Reconstruction of an Extended Regulatory Network Incorporating Regulations Mediated by Small RNA and Protein–Protein Interactions

**DOI:** 10.3390/microorganisms9071395

**Published:** 2021-06-28

**Authors:** Juan M. Escorcia-Rodríguez, Andreas Tauch, Julio A. Freyre-González

**Affiliations:** 1Regulatory Systems Biology Research Group, Laboratory of Systems and Synthetic Biology, Center for Genomic Sciences, Universidad Nacional Autónoma de México, Av. Universidad s/n, Col. Chamilpa, Cuernavaca 62210, Morelos, Mexico; escorcia@ccg.unam.mx; 2Centrum für Biotechnologie (CeBiTec), Universität Bielefeld, Universitätsstraße 27, 33615 Bielefeld, Germany; tauch@cebitec.uni-bielefeld.de

**Keywords:** *Corynebacterium glutamicum*, regulatory interactions, regulatory network, curation, network inference, systems, modules, NDA, regulogs

## Abstract

*Corynebacterium glutamicum* is a Gram-positive bacterium found in soil where the condition changes demand plasticity of the regulatory machinery. The study of such machinery at the global scale has been challenged by the lack of data integration. Here, we report three regulatory network models for *C. glutamicum*: *strong* (3040 interactions) constructed solely with regulations previously supported by directed experiments; *all evidence* (4665 interactions) containing the *strong* network, regulations previously supported by nondirected experiments, and protein–protein interactions with a direct effect on gene transcription; *sRNA* (5222 interactions) containing the *all evidence* network and sRNA-mediated regulations. Compared to the previous version (2018), the *strong* and *all evidence* networks increased by 75 and 1225 interactions, respectively. We analyzed the system-level components of the three networks to identify how they differ and compared their structures against those for the networks of more than 40 species. The inclusion of the sRNA-mediated regulations changed the proportions of the system-level components and increased the number of modules but decreased their size. The *C. glutamicum* regulatory structure contrasted with other bacterial regulatory networks. Finally, we used the *strong* networks of three model organisms to provide insights and future directions of the *C.*
*glutamicum* regulatory network characterization.

## 1. Introduction

*Corynebacterium glutamicum* is a Gram-positive soil bacterium, industrially relevant due to its amino acid production proficiency. It is also a model organism for the study of regulatory networks [1], along with other bacteria such as *Escherichia coli*, *Bacillus subtilis*, and *Streptomyces coelicolor*. These model organisms are usually compared, and diverse differences have been found (e.g., while *C. glutamicum* grows by apical elongation, *B. subtilis* and *E. coli* grow by lateral elongation [2]). Some aspects of the transcriptional regulatory mechanism of *C. glutamicum* have also been found to be different from those in other model organisms [3]. In contrast to *E. coli*, repression is the most common regulatory mechanism in *C. glutamicum* [4], and unlike *B. subtilis* and *E. coli*, which have diauxic growth due to the preferential consumption of one carbon source over others, *C. glutamicum* cometabolizes glucose with several other carbon sources [3]. In terms of σ factors, *E. coli* and *C. glutamicum* have seven, while *B. subtilis* has 17, and over 60 σ factors have been found in the *Streptomyces* species [5].

One of the challenges for the study of their transcriptional machinery at the global scale is the lack of data integration and the incompleteness of their global regulatory networks despite being model organisms [1]. The network incompleteness situation is worst for nonmodel organisms for which little or none is known about their transcriptional machinery. Even though high-throughput technologies speed up the reconstruction of regulatory networks, network models reconstructed solely with high throughput experiments present unusual structural properties when compared with other reconstructions performed mainly by conventional experiments (e.g., lower clustering coefficient [6]). Moreover, the number of sequenced genomes scales rapidly, especially for bacteria, so that even with high throughput experiments, we cannot cope with all of them. Computational approaches for the inference of regulatory networks based on gene expression data are still emerging. Proof of that is their modest performance for model organisms in the DREAM5 challenge [7] and the inconsistency between gene expression data and the model used for regulatory networks [8], although a reassessment with more complete networks and a larger number of model organisms is required [1]. An integrative approach of expression data and regulatory binding sites have shown to improve the prediction, but most of that improvement is by the binding sites approach, which provides more biological information (e.g., [8]).

When inferring regulatory interactions with transcription factor (TF) binding sites data, the approaches can be classified into three major groups: phylogenetic footprinting, regulon expansion, and regulatory interaction transfer. The latter two approaches require previous regulatory information to increase the target genes (TGs) for the TFs in a network or transfer the regulatory information between organisms, respectively. On the other hand, phylogenetic footprinting does not require previous regulatory information but is limited to the identification of coregulated genes by a common TF. However, when the cognate regulator is unknown, its identification is not trivial [9,10] due to the small size of the regulatory sequences and their overlap for some close homologous proteins. The transfer of regulatory interactions can be directly through the orthology of both TF and TG conservation or by filtering for TF binding sites in the promoter region of the TG (also known as a regulog analysis [11]). The latter provides the best results, helping us to reduce spurious interactions that are not conserved in the organism of interest [12].

Previously, we studied the functional architecture of the *C. glutamicum* regulatory network with regulations by TFs binding to DNA acting at the level of transcription initiation (transcriptional regulatory network) and compared its connectivity distribution to those in *E. coli* and *B. subtilis* regulatory networks [13]. Since then, a plethora of studies has continued unveiling novel transcriptional regulatory mechanisms in *C. glutamicum*. However, the study of the regulatory mechanisms has not been restricted to TF–DNA interactions. Some protein–protein interactions (PPis) are directly involved in transcription regulation (e.g., adenylated GlnK binding to AmtR (repressor) to release it from the DNA). Additionally, the inclusion of post-transcriptional regulations mediated by sRNAs into global regulatory networks has been performed in other organisms (e.g., [14] in *E. coli* as an undirected network). Previous versions of the *C. glutamicum* transcriptional regulatory network have been used for the transfer of regulatory interactions to other corynebacterial strains hosted in the CoryneRegNet database [15], the construction of a model for the inference of the number of interactions once the regulatory networks are complete [6], for an assessment of the NDA robustness to random remotion of nodes and interactions [13], as the gold standard for the benchmarking of a network inference approach based on sequence data (unpublished results), and as a reference for the identification of global regulators [16].

Here, we update the two previous transcriptional regulatory network models for *C. glutamicum* (Abasy IDs: 196627_v2018_s17 and 196627_v2018_s17_eStrong; hereinafter referred to as *all evidence* and *strong*, respectively) with hundreds of curated TF–DNA interactions, their effect, and their corresponding confidence level. In the *all evidence* network, we also included curated PPi that have a direct effect on gene transcription, such as anti-σ–σ factor interactions and the formation of heteromeric regulatory complexes. We incorporated interactions mediated by regulatory small RNAs acting at the post-transcriptional level in a third network model (hereinafter referred to as *sRNA)*. We deposited all three network models in the new v2.4 of Abasy Atlas. Our continuous curation of the *C. glutamicum* regulatory network has produced a set of five historical snapshots that, together, recount the curation process that has spanned 11 years. These historical snapshots are also available in Abasy Atlas.

After this update, *C. glutamicum* moves from the fourth to the second position among the organisms with the most complete regulatory network in Abasy Atlas, according to our recently published model of the total number of interactions a complete regulatory network has [6]. We discuss the global structural properties of the three network models in the context of the previous versions of the transcriptional regulatory models and more than 40 other bacterial networks from Abasy Atlas, the most complete collection of experimentally validated regulatory networks [1]. We analyzed the organizing principles and the system-level components of the three networks to identify the effects of the inclusion of interactions supported by nonstrong experiments, protein–protein interactions, and post-transcriptional layer regulation by sRNAs. Finally, we use strongly supported regulatory networks from *S. coelicolor*, *B. subtilis*, and *E. coli* to gain knowledge of the DNA-binding TFs for which no TGs have been characterized in *C. glutamicum*, and we provide a list of potential interactions retrieved through a strict and conservative computational pipeline using the most precise tools to identify regulations.

### A Primer on Analyzing Regulatory Networks

The concepts and procedures used in the field of network biology have been summarized and explained in-depth and with great clarity in previous works [17,18,19,20,21]. Nevertheless, in this section, we summarize the state of the knowledge and main concepts required to analyze the relationship between the structure and function of regulatory networks.

The abstraction of a regulatory network can be represented as a group of nodes and directed arcs. The nodes represent the entities of the network (commonly genes or sRNAs), and the arcs represent the direction of the interaction between two nodes. For example, the requirement of GlxR for the transcription of *ramA* can be represented as *glxR* → *ramA*, while a negative effect (such as GlxR on *acnR* transcription) is usually represented as *glxR* ⊣ *acnR*. We use the gene symbol (or locus tag in the case no name has been assigned yet) to consistently represent the sequence of the interactions, for example, *sigA* → *glxR* ⊣ *acnR*, and so on (the housekeeping σ factor is required for the transcription of *glxR*, and GlxR hinders the transcription of *acnR*). Nodes representing other biological entities can also be included in the network. The *C. glutamicum* networks herein reported contain three types of nodes: genes, heteromeric protein complexes, and sRNAs. Heteromeric protein complexes are conformed by two or more regulatory proteins transcribed by different genes and are included in the network to reduce redundancy and improve representation accuracy [1]. The effect of the sRNA regulatory interactions is carried out at the post-transcriptional level. These interactions are included in the networks with an sRNA label in their corresponding Abasy ID [1]. The importance of the inclusion of sRNAs in bacterial regulatory networks is relatively recent [14], and there is little information regarding these types of interactions in bacterial regulatory networks.

Once the interactions are merged to form a global regulatory network, we can compute the **connectivity degree** of the nodes (k), which represents the number of interactions of a node with the rest of the network, regardless of the direction. In some scenarios, the connectivity degree can be more informative if the direction of the interactions is considered. The out-degree (**k_out_**) of a node is the number of nodes it regulates. The nodes with a k_out_ greater than zero are defined as regulators. The k_out_ is the most applied connectivity in regulatory networks (e.g., for the identification of proteins required for the transcription of a large fraction of the network: global regulators). The in-degree (**k_in_**) is the number of regulators involved in the transcription of a given gene/sRNA. An exception is the incoming interactions in heteromeric complexes that represent the formation of the complex instead of their regulation, despite that the relationship is causal, as the presence of the subunits is required to produce the heteromeric complex. Hence, the heteromeric complexes have incoming interactions only from the subunits required for its conformation, while the subunits have outgoing interactions only to the heteromeric complexes they are part of [1]. These types of interactions are underrepresented in the network and, therefore, not specified in most cases. **k_max_** is defined as the largest connectivity value of the network and equals the k_out_ of the global regulator with the largest set of TGs. The **auto-regulations** represent a direct transcriptional effect of the regulator onto its own coding sequence.

The average clustering coefficient quantifies the modularity of a network. This structural property is an example where the direction of the interactions is disregarded, as modularity is defined as the degree to which the components of a system are separated or combined. The **clustering coefficient** of a node is defined as the fraction of its neighboring nodes that are connected to each other, relative to the potential interactions that could exist among them. For example, node A, having as neighbors only the nodes B and C, will have a clustering coefficient of one if an interaction exists between B and C (regardless of the direction of the interaction) because the potential number of interactions between the neighbors of A is only one. The clustering coefficient of A is zero if there is no interaction between B and C. Once the clustering coefficient is calculated for every node having at least two neighbors in the network, the values are averaged. For an illustrated example, please see Box 1 in reference [19]. **C(k)** shows a distribution of the average clustering coefficient for the nodes with connectivity k. Similarly, the distribution of the connectivity of the nodes is denoted as **P(k)**, provided by the probability of a node having k interactions. It has been previously debated whether the P(k) of real networks is truly governed by a power-law distribution, where a few nodes have most of the interactions [22]. Recently, using several statistic methods, we demonstrated that regulatory networks truly follow a power-law distribution—they fit other power-law-like distributions better than a Poisson distribution, regardless of the completeness of the network—and that the sole coefficient of determination (R^2^) is a good proxy to assess the goodness-of-fit of the model [6].

A network component is a group of nodes in which every pair is connected by at least one path. Regulatory networks do not always comprise a single component. Commonly, small groups of nodes can be isolated from the rest of the network. This is frequently observed in nonmodel organisms for which only some groups of nodes have been studied. Whether regulatory networks are truly multicomponent, or this is only a consequence of network incompleteness, is still an open question. The **giant component** is the largest component of the network, and its size is determined by the number of nodes it covers. In regulatory networks, the global TFs, such *sigA*, increase the fraction of nodes in the giant component. The higher the fraction of nodes in the giant component, the more cohesive the network is. The giant component of a network is the representative part of the network for most structural properties such as density. **Network density** is the fraction of interactions from the fully connected network (where every node would have a directed interaction to itself and every other node in the network) that exists in the actual network. The detection of a constrained space for density values in bacterial regulatory networks [6] allowed us to infer the number of interactions expected once the curation of the network is completed [1] in order to identify some differences in the curation state of the regulatory networks.

Most of the definitions mentioned before are applied to the **κ-value** (Κappa value), which is defined as the point of the C(k_out_) distribution where the change in normalized k_out_ connectivity equals the change in the clustering but with the opposite sign. The κ-value is used as a threshold for the identification of global regulators and has shown high precision and sensitivity to different bacterial regulatory networks such as *E. coli* [23], *B. subtilis* [24], and *S. coelicolor* (unpublished results) while being conservative (high precision, low sensitivity) on an earlier version of the *C. glutamicum* regulatory network [13].

The global regulators shape the highest hierarchy in the diamond-shaped structure unveiled by the natural decomposition approach (NDA). The **NDA** is an in-silico technique that deconstructs a regulatory network to naturally identify its structure and reconstructs it with the nodes classified into one of four classes: global regulators (**GRs**), modular nodes (**Mds**), intermodular nodes (**IMs**), and basal machinery (**BM**). Global regulators are the TFs with a low clustering coefficient and a k_out_ greater than the κ-value. Once the GRs have been identified, the BM is also unveiled as the TGs that are regulated only by GRs. The direct GR–BM regulation is required for fast responses without previous modulation of intermediates. GR and BM nodes and their interactions are removed from the network as well as the rest of the nodes with k_out_ = 0 (putative structural genes). The remotion of these structural genes will lead to isolated groups of Mds (modules) that work together for a common purpose. Finally, the structural genes are reinserted into the network, preserving their original interactions, and they are included into the module of their regulators only if all their regulators are from the same module. Otherwise, they are included as IMs, integrating the signals from different modules. For further details about the NDA methodology, please see Figures 1 and 2 in [13], where the NDA is described and applied to an earlier version of the *C. glutamicum* transcriptional regulatory network. Noteworthy, this diamond-shaped hierarchy has been found to be structurally conserved even between phylogenetically distant organisms [24]. The NDA classification is robust to random remotion of interactions and nodes [13], but the curation state of the network can alter the class of some nodes. This applies mainly to the IM and BM nodes that can be included in the Md class in a later (more complete) version of the network.

## 2. Materials and Methods

### 2.1. Curation and Network Definition

Four types of interactions were defined for consideration in this new version of the *C. glutamicum* networks: (1) homomeric-TF–DNA comprehending interactions between DNA-binding TFs (including σ factors) and the DNA, altering the gene expression; (2) sRNA–RNA interactions, occurring at the post-transcriptional level, modulating the concentration of the proteins; (3) protein–protein interactions class 1 (PPi-cI), defined as PPis with a causal regulatory effect, such as anti-σ–σ interactions; (4) PPi class 2 (PPi-cII), a form of TF–DNA interaction where the TF is a heteromeric protein complex its with cognate subunits—complex interactions. Two levels of confidence are defined for the interactions: strong, if the interaction is supported by a TF–DNA direct binding experiment (e.g., footprinting with purified protein), and weak, otherwise. Even though other types of interactions considered for this version might be supported by a direct experiment (e.g., yeast two-hybrid assay for PPi-cI), we only included homomeric-TF–DNA and heteromeric-TF–DNA interactions (PPi-cII) in the *strong* network. The *all evidence* network includes interactions supported by any experimental evidence, keeping the label “strong” only for those interactions taken from the *strong* network. For the *all evidence* network, all but the sRNA-mediated regulations are considered, while the *sRNA* network includes every type of interaction regardless of the experiment supporting it. The three networks reconstructed in this work have been deposited in the new v2.4 of Abasy Atlas.

The curation of strong interactions was carried out manually by screening the PubMed library for publications describing regulatory interactions of *C. glutamicum*. Interactions are classified as strong when the respective paper contains experimental evidence of a TF–DNA interaction. In most cases, the TF of interest is purified and its direct interaction with DNA is demonstrated in vitro. Approaches like this also lead to the experimental identification of the DNA binding site sequence. For the recovery of weakly supported interactions, we reviewed the literature to identify TGs for the TFs already present in the *all evidence* network. We used as keywords “glutamicum”, “regulon”, ”target genes”, and the name symbol of the gene or its locus tag. Then, we followed a set of rules to include the interactions for every TF–TG pair of nodes: (1) an interaction does not exist in the network unless it is already in the previous version; (2) an interaction that is not part of the previous version does not exist unless there is experimental evidence to support the interaction; (3) an interaction supported solely by computational predictions is not included in any of the networks; (4) an interaction weakly supported by an experiment is part of the network until contradictory evidence is found (e.g., gene overexpression supported by microarrays data but invalidated by RT-PCR).

We included in the *sRNA* network the regulatory interactions by anti-sense sRNAs from reference [25]. The authors included as anti-sense sRNA every sRNA that is transcribed in the opposite strand of a gene, starting within 100 nt of the 5′-end of an opposite CDS or within 60 nt from the 3′-end of an opposite CDS [25]. The authors identified two other types of sRNAs, but regulatory interactions were only assigned to anti-sense sRNAs. For the name of the sRNAs, we used the nomenclature suggested by the authors—*cgb_xxxxx*—to ease the identification of the nodes representing sRNAs in the *sRNA* network. The effect of the interactions was set to unknown—“?”—and most of the sRNAs regulate the gene transcribed in the opposite DNA strand. We included the sRNAs as independent nodes. We acknowledge that this artificially increases the genomic coverage for the *sRNA* network (counting twice the genes with an asRNA). However, assigning the interaction to the coding gene would be misleading and would inflate the number of self-loops in the network even when the sRNAs might be transcribed through its own promoter. As previously discussed, interaction coverage is a better proxy for network completeness than genomic coverage [6]. Although the authors provide the σ factors required for the transcription of the sRNAs, we did not include these σ-DNA interactions as they were solely supported by DNA-binding motif computational predictions and we have identified a high number of false-positives in the search for binding sites for σ factors. Moreover, interactions supported solely by computational predictions are not considered for Abasy Atlas networks [1]. Interactions involving a protein-coding gene not mapping to a cgl-number or from another strain are not included in the networks but collected in a separated file (Appendix A).

### 2.2. Genome Annotation and Upstream Sequences

Genome annotations used in this work were retrieved from NCBI [26] for the following organisms (accession code and version): *Corynebacterium glutamicum* ATCC 13032 (NC_006958.1), *Streptomyces coelicolor* A3(2) (NC_003888.3), *Bacillus subtilis* subsp. subtilis str. 168 (NC_000964.3), and *Escherichia coli* str. K-12 substr. MG1655 (NC_000913.3). Upstream (up to −300 to +50) sequences with reference to the translation-start codon, for the four genomes, were retrieved from the RSAT suite [27] with the retrieve-seq tool, preventing overlap with neighboring genes.

### 2.3. Regulatory Networks for Other Organisms

All the regulatory networks used in this work were downloaded from Abasy Atlas, a large collection of manually curated transcriptional regulatory networks [1]. The set of nonredundant networks is defined as the most recent regulatory networks for each organism available in Abasy Atlas, resulting in a dataset of 42 regulatory networks for 42 bacterial strains. When using the nonredundant set as a background for the herein reported regulatory networks of *C. glutamicum*, the set includes the regulatory networks of all other organisms (41) plus the three herein reported networks.

### 2.4. System-Level Components

Nodes were classified into one of the four system-level component classes: GRs, BM, Mds, and IMs were retrieved from Abasy Atlas. The classification of the nodes has been previously described [13]. In the following paragraph, we briefly describe the NDA, the approach used for the classification of the nodes and module identification: The κ-value is computed for the identification of GRs. Every node with a number of directly regulated TGs greater than the κ-value is classified as a GR and removed from the network, along with their interactions. The remotion of the global regulator nodes leaves some nodes isolated. The isolated nodes that are solely regulated by global regulators are classified as BM, representing structural components required for elemental functions such as the subunits for RNA core polymerase. The nodes with no regulated genes in the remaining network are labeled as structural nodes and removed in order to identify an isolated group of nodes (modules) to be classified as Mds. The nodes labeled as structural are reintegrated to the network as part of a module if all of their regulators belong to the same module; otherwise, they are labeled as IM components, which integrate the signals from two or more modules responding to different conditions.

### 2.5. Comparison of Nodes and Interactions of C. glutamicum with Other Bacterial Regulatory Networks

To quantify the fraction of *strong* interactions in each network, we computed the ratio of regulatory interactions classified as *strong* in each of the *all evidence* regulatory networks deposited in Abasy Atlas, including the *all evidence C. glutamicum* network herein reported, and plotted the distribution. We reconstructed the previously reported model, developed to predict the size of regulatory networks [1], by using an expanded dataset including the herein reported *C. glutamicum* regulatory networks and robust linear regression. We then reassessed the goodness-of-fit of the model by recomputing the adjusted coefficient of determination. Regulatory networks of *C. glutamicum* were highlighted in the distributions to ease identification and comparison with previous versions.

### 2.6. Global Structural Properties

All the structural properties reported in this work were retrieved from Abasy Atlas [1] version 2.4. For comparison with other bacteria, the values reported were normalized as follows: The number of autoregulations was normalized by the number of regulatory nodes (those with the potential to have an autoregulation). To ease the comparison of density values in a plot, each of them was multiplied by 10. Please note that this modification is used only to compare the properties. The k_max_ was normalized by the number of nodes in the network (potential targets). The κ-value was normalized by the k_max_. The size of the giant component was normalized by the number of nodes in the network. No normalization was applied to compare the *C. glutamicum* network across versions and evidence levels. Instead, we used a log2-fold change ratio of the properties’ value relative to the corresponding value for the earliest network in the case of different versions and the smallest network in the case of comparing different evidence levels.

### 2.7. System-Level Components

Node classification, module identification, and their annotation were retrieved from Abasy Atlas [1] version 2.4. For the graphic representation of node classification, the values were computed using a log10 scale. For the representation of module size, actual values were used for the treemapping plot. For distribution of the number of modules, the nonredundant set of regulatory networks from Abasy Atlas version 2.4 was used, and the herein reported networks were highlighted and labeled to ease identification. For the comparison of the nodes in each NDA class for the three networks reported here, we used the Simpson similarity index, defined as the number of common elements between two sets divided by the minimum of the two numbers. Hence, the similarity index can take values from zero (no overlap at all between the two sets) to one (one set is a subset of the other). For the interactions from GRs and Mds to the four classes, we computed the fraction of interactions between each class, ignoring interactions from BM and IM classes (less than 1% of the network), which are attributed to missing interactions that will be included in the future curation of the network (e.g., *cgb_20925* regulating *sigA*). Matplotlib, Seaborn, Numpy, and Squarify libraries from Python were used to compute and plot the results.

### 2.8. Regulog Analysis

For the selection of source organisms, we used the last *strong* version of those organisms having strong regulatory networks, namely, *Escherichia coli* K-12 MG1655 (Abasy ID: 511145_v2020_sRDB18-13_eStrong), *Bacillus subtilis* strain 168 (Abasy ID: 224308_v2008_sDBTBS08_eStrong), and a curated *Streptomyces coelicolor* network, with curated strong interactions until 2019 (unreported network). Regulog analysis is based on the premise that regulatory sites are more conserved than the rest of the noncoding sequences because they are required for the cell to survive. Given the basis of the approach, the best strategy is to use phylogenetically closely related organisms [11,28]. Unfortunately, model organisms for which a *strong* regulatory network is available are phylogenetically far from each other, but we still can use them to study essential, conserved interactions [24]. The closest model organism with a highly complete regulatory network is *Mycobacterium tuberculosis* (Abasy ID: 83332_v2018_s11-12-15-16), but its regulog analysis has been previously used to transfer interactions in the opposite direction (from *C. glutamicum* to *M. tuberculosis*) [29], and the remaining interactions are mostly supported by weak evidence.

For the identification of orthologous genes, we used the OMA standalone [30] with genome sequences from NCBI (see above). We used the OMA classification of orthology relationship type and kept only the one-to-one orthology relationships. To construct the position weight matrices, we used MEME [31], Bioprospector [32], and MDscan [33] with the upstream sequence of TGs for each TF with at least one *strong* evidence supporting the interaction. Upstream sequences were defined as up-to −300 to +50 bp, relative to the translation-start codon. Then, we used FIMO [34] to find individual matches of the matrices in the upstream sequences of the complete set of *C. glutamicum* one-to-one orthologous genes using a *p*-value of 1 × 10^−4^ as a threshold to form TF–TG putative interactions. Gene identifiers for the TFs and TGs were mapped to the *C. glutamicum* genome annotation, and the interactions obtained with each of the three motif-finding tools were integrated by a vote-counting approach, which has been found to improve predictions [7], prioritizing the interactions considered as “more reliable” by the three motif-finding tools.

## 3. Results and Discussion

### 3.1. The Regulatory Networks of C. glutamicum and Potential Applications

In this section, we report the new regulatory network models of *C. glutamicum*, their differences, and the statistics comparing them with the previous version and discuss some potential applications of our network models. We reconstructed three regulatory network models: (1) The *strong* network (Abasy ID: 196627_v2020_s21_eStrong), conformed solely by DNA-binding TFs—mediated interactions that are supported by a direct experiment (e.g., footprinting with purified protein); (2) The *all evidence* network (Abasy ID: 196627_v2020_s21), conformed by every type of interaction at the transcriptional level that is supported by any experimental evidence and not discarded by any other; (3) The *sRNA* network (Abasy ID: 196627_v2020_s21_dsRNA), containing the *all evidence* network plus 545 post-transcriptional interactions mediated by regulatory sRNAs (Figure 1). The *strong* network is a subset of the *all evidence* network, while the *all evidence* network is a subset of the *sRNA* network (Appendix A). We deposited the three reconstructed networks in the new v2.4 of Abasy Atlas, each of them providing a different level of completeness (Appendix A) that is useful in different scenarios. For example, even though the *strong* network is the smallest one, the confidence level of its interactions makes this network the best alternative to be used as the gold standard for benchmarking approaches for the inference of directed regulatory networks (such as those based on regulatory binding sites). On the other hand, benchmarking of network inference tools based on transcriptomic data might tend to be penalized when using only the *strong* network, as it only contains direct TF–DNA interactions that cannot accurately be predicted based solely on transcriptomic data [8]. In that case, the *all evidence* network can be used as the gold standard, as it includes a broader scope of experimentally supported interactions that have not been reported as spurious. The *sRNA* network is the most comprehensive and, therefore, the best suited to study the biological regulatory mechanisms of *C. glutamicum*. Having reliable regulatory network models has proven to be important even for synthetic biology, for example, to engineer resource allocation by rationally modifying the transcriptional regulatory network [35].

### 3.2. Global Networks of C. glutamicum Are Quite Different from other Bacterial Networks in Terms of Their Structural Properties

In this section, we analyze the global structural properties of the *C. glutamicum* regulatory networks in the context of the whole Abasy Atlas dataset. Previously, our group found a constrained complexity in the regulatory networks [6] and leveraged it to create a model for the inference of the size of regulatory interactions expected once network curation is complete [1]. We identified a few networks falling outside of the prediction area (see Figure 5 in reference [1]), *C. glutamicum* being one of those organisms, namely, for the later versions containing the sigmulons of the housekeeping σ factor *sigA*. We found that this was a result of a low number of weakly supported interactions in contrast with other bacterial regulatory networks (Figure 2a), mainly because the *C. glutamicum* regulatory network has been highly curated in-house, giving preference to strongly supported interactions and resulting in an overrepresentation of these interactions in contrast to other bacterial regulatory networks. The inclusion of weakly supported interactions better fits the *C. glutamicum* network into the model (Figure 2b). Note that the *strong* version of the network follows the model poorly as *sigA* directly regulates 85% of the network nodes. Moreover, the fit of the *all evidence* network is affected by the inclusion of sRNA-mediated interactions (RNA in Figure 2b). This is a result of many sRNAs regulating only one gene in most cases.

Related to this, we expect the average clustering coefficient to decrease as the node/interaction ratio increases. The clustering coefficient of a node in the network is determined by the fraction of neighbors connected to each other. As expected, the average clustering coefficient of the *all evidence* network is higher than the other two networks of the same time frame (network version) (Figure 2c) as it exhibits a better equilibrium (closest to 1) of the genomic/interaction coverage ratio (Appendix A). Interestingly, despite the *C. glutamicum* networks exhibiting a higher node/interaction ratio, they have a higher clustering coefficient than most of the bacterial regulatory networks (Figure 2c), perhaps because of a higher level of curation of the organism due to its biotechnological relevance. The density of the *C. glutamicum* networks is slightly lower than the rest of the bacterial regulatory networks. However, note that this difference is so small that even a 10-time magnification of the variance of density values is very small (Figure 2c). This is expected due to the constraint governing the complexity of regulatory networks [6].

The fraction of nodes acting as transcriptional regulators is constrained in bacteria, beyond considering only the DNA-binding TFs (Figure 2c). The *C. glutamicum* regulatory network models show a different behavior; while the network including sRNA-mediated interactions falls on the upper boundary (~25%), the other two networks fall on the lower boundary of the distribution (5%), even when the latter includes most of the DNA-binding TFs of *C. glutamicum*. For most organisms, the k_max_ is below 50% of the nodes in the network. However, the regulatory networks for *C. glutamicum* are outliers in the distribution (Figure 2c) due to the *sigA* interactions. The size of the giant component can be represented by the fraction of the network it comprehends. For most regulatory networks, this fraction is close to one (Figure 2c), especially in the case of *C. glutamicum*, whose networks with no sRNA regulation are practically a single component, showing the cohesiveness of these networks.

The κ-value is the threshold to identify global regulators. Every network has a different κ-value that relies on its hubness and modularity, but larger k_max_ values result in larger κ-values. To make the κ -values comparable, we normalized them by the k_max_ of the cognate network, allowing κ to take values between 0 and 1. Interestingly, the normalized κ-value seems to be also constrained to values lower than 0.25, and the values for the three networks of *C. glutamicum* are overlapped. This suggests that the κ-value is robust to the inclusion of weakly supported interactions and sRNAs. Moreover, this agrees with previous analysis on the robustness of the inference of global regulators to random removal of nodes and interactions [13]. However, in-depth studies with other sampling approaches and other organisms are required. Autoregulations in a regulatory network allow mechanisms to modulate themselves. A higher number of autoregulations in the networks provide a faster response of the organism to the changing conditions [37]. *C. glutamicum* requires the adaptation to different media conditions in the soil; therefore, a high number of autoregulations is expected (Figure 2c–f), where the *strong* and *all evidence* networks are above most regulatory networks. However, the fraction of autoregulations in the network containing sRNA-mediated interactions is much lower because of the large number of regulatory sRNAs that bind to other RNA but not to themselves.

### 3.3. System-Level Components of the C. glutamicum Regulatory Networks

The regulation of gene transcription is organized into different hierarchical layers. Previously, we have described a large-scale modeling approach to characterize the nodes of a regulatory network: the NDA (natural decomposition approach). The NDA classifies each node of the network into one of four system-level components: GRs, BM, Mds, and IMs. Regulatory networks having a diamond-shaped hierarchy have been found in different bacteria such as *E. coli* [23,24], *B. subtilis* [24], and a previous version of the *C. glutamicum* transcriptional regulatory network [13]. The hierarchy is divided into three layers (Figure 3a): the top layer, composed solely of global regulators (coordination layer), is the smallest one and can directly regulate the four NDA classes; the middle layer (processing) is composed of Mds and BM, the two largest NDA components, both regulated by the coordination layer, but with only the Md class providing feedback to the top layer (i.e., some Md TFs regulate GRs); the last layer (integration) assimilates the combinatorial disparate signals provided by GRs and Md TFs belonging to different modules into a single coordinated response, essential to adapting to environmental changes.

Using the *all evidence* network as an example, the coordination layer is composed of nine GRs (Figure 3a). As expected, the first GR, when sorted by their K_out_, is the housekeeping σ factor (*sigA*), required for the transcription of 85% of the nodes in the network. It is followed by the dual regulator *hrrA*, involved in the transcription of 21% of the network. The rest of the global regulators (and their corresponding rounded regulated network percentage) are *ramA* (11%), *glxR* (8%), *sigH* (6%), *ramB* (5%), *atlR* (4%), *mcbR* (4%), and *dtxR* (3%). The difference in regulated genes by the first and second global regulators is enormous, and this gap becomes smaller for the rest of the TFs. This is what provides the hierarchical structure to the network fitting a power-law distribution (a small fraction of nodes has most of the interactions). More than 66% of the *all evidence* network nodes are classified as BM. Examples of BM are the *rpoA*, *rpoB*, *rpoC*, and *rpoZ,* genes coding for RNA polymerase subunits.

Please note that the BM class is composed of nonregulators and is inferred based on their regulation solely by GRs. Therefore, some of its members can be transferred to the Md or IM class if they are found to be regulated by a TF from the Md class. However, it is very unlikely for a structural gene belonging to the Md class to become part of the BM (because it requires losing regulations mediated by an Md TF) and even less likely for IMs because it would require the loss of at least two Md-mediated interactions. For these reasons, a regulatory network with high genomic coverage tends only to reduce the BM as more interactions are included. On the other hand, regulatory networks with low genomic coverage are highly likely to be lacking interactions by GRs and their BM will increase with genomic coverage. It was the case for the large increase in genomic coverage in a previous update of the *C. glutamicum* transcriptional regulatory network from 2011 (genomic coverage: ~24%) to 2016 (genomic coverage: ~71%), which was mainly due to the inclusion of the *sigA* sigmulon, causing an increment of BM from 60% to 77% of the network. The Md class is composed of ~28% (691/2441) of the network, divided into locally independent modules (see below). Finally, the IM class is composed of ~5% (117/2441) of the genes in the network, all of them being structural genes (nonregulators with k_out_ = 0).

The Md class is further divided into locally independent modules, groups of genes that are combinatorially expressed in response to specific media conditions. In the case of the *all evidence* network, the Md class is divided into 64 modules, 18 of them (28%) enriched with one or more biological functions (Figure 3b). We used a “guild-by-association” approach to assign a biological function to nodes that have no previous annotation due to poorly annotated orthologs but belong to enriched functional modules (e.g., a module where all but one node has a GO annotation for DNA repair) [38]. The proportions for each NDA class are conserved in the network containing only strongly supported interactions, BM being the largest class, followed by Mds, IMs, and lastly, GRs. On the other hand, when regulations mediated by sRNAs are integrated (*sRNA* network) to the *all evidence* network, the proportions change for the BM and Md classes, the Md class being the largest one (Figure 3c). The number of modules is largely increased with the inclusion of sRNA regulations (Figure 3e), being an outlier in the distribution of the number of modules of bacterial regulatory networks, while the *strong* and *all evidence* networks have similar values. Even though the *sRNA* regulatory network is larger (Figure 1) and every sRNA but *cgb_20925* is included in the Md class, this does not compensate for the number of modules in the network. This is observed when we compare the distribution of the size of the modules in the networks (Figure 3e). This is also a result of the sRNAs regulating many of the nodes that are solely regulated by *sigA* in the *all evidence* network, transferring them from the BM class to the Md class and decreasing the BM class from 66.5% to 44.2% of the network.

Comparison of the size of the classes provides insights into their differences and similarities; contrasting the elements of each class contributes more to the comparative purpose. We used the Simpson similarity index to identify the overlap of two classes, taking as reference the smallest one in each comparison. Thus, the Simpson similarity index for two sets, one being a subset of the other, is 1. On the other hand, two sets having no overlap at all have an index of 0, and two sets where half of the smallest one is a subset of the largest one will have 0.5 as an index. For each NDA class, we computed the Simpson similarity index for every pair of networks and found that the *all evidence* and *sRNA* networks are more similar to each other than to the *strong* network (Figure 3d). This is expected since the *all evidence* network is a subset of the *sRNA* network (Appendix A). Please note that even though one network is a subset of the other, NDA classification is performed independently for each network; therefore, the class of a node can change from one network to another. Previous analysis of the robustness of the NDA classifications to random remotion of nodes and interactions showed the IM class is the least conserved class [13]. Surprisingly, this was not the case in the class conservation across network models, where the Md class was the least conserved (Figure 3d). This was caused by the inclusion of sRNAs in the Md class. On the other hand, the similarity index of the IM class between the *all evidence* and *sRNA* networks was not affected because even though the number of intermodular nodes increased (from 117 to 194), one is a subset of the other. Consistent with the previous robustness analysis of the *C. glutamicum* network to random interactions remotion [13], the basal machinery is well conserved, while the GR class is the most conserved class, with a similarity index of 1 for the three comparisons between the networks. This is because the *all evidence* and *sRNA* networks have the same global regulators (listed in Figure 3a), and the *strong* network has four of these nine global regulators (*sigA*, *sigH*, *dtxR*, and *glxR*).

When analyzing the communication between classes (Figure 3f), most of the interactions in the network occur from GR → BM, followed by GR → Md and Md → Md (regulations between modular TFs). For the *sRNA* network, GR → BM is decreased, while the Md → Md interactions are increased due to the inclusion of sRNAs in the Md class, regulating nodes that used to be part of the BM but are now included in the Md class. The GR and IM classes have virtually the same fraction of regulations coming from GRs and Md TFs in the *C. glutamicum* network, but further investigation in other organisms is required to assess the conservation of the proportions.

### 3.4. Recovering Conserved Interactions from Other Model Organisms

Regulog analysis is based on the premise that a TF–TG interaction from organism A is conserved in organism B if B has an ortholog of the TF, an ortholog of the TG, and a binding site for the TF in the promoter region of TGs [11]. As regulatory networks are highly plastic, a caveat of the regulog analysis is the functional divergence of one of the components involved in the interaction, especially for the TF [39]. Therefore, this analysis is usually applied between phylogenetically closely related organisms and is useful to transfer interactions from one model organism to others, for example, from *C. glutamicum* to other Corynebacteriales [15]. However, model organisms for the study of regulatory networks are phylogenetically far from each other, which allows the transfer of interactions from model organisms across several bacterial genera [10,15,40]. We restricted our source organisms to purely *strong* networks as they only contain directed TF–DNA interactions supported by at least *strong* evidence, namely, *E. coli*, *B. subtilis*, and *S. coelicolor*. Please note that despite the high completeness of the network for *M. tuberculosis* [1] and the closeness to *C. glutamicum* in the phylogeny, we did not use this network as a source since it was mainly constructed using only high-throughput technologies without further confirmation with directed experiments. This causes an unusually lower clustering coefficient for the network (see Figure 5 in reference [6]). Moreover, *C. glutamicum* has been used as a source organism for the inference of regulatory interactions in *M. tuberculosis* [41]. We acknowledge the caveats of using distant organisms for regulog analysis; for this reason, we applied strict conditions during the entire workflow, prioritizing precision at the expense of losing many potential interactions.

Using *S. coelicolor, B. subtilis*, and *E. coli* as source organisms (Figure 4a), we aimed to identify conserved interactions despite their phylogenetic distance (especially for *B. subtilis* and *E. coli*). To do so, first, we identified the pair-wise genome-wide orthologs between the source organisms and *E. coli* with the OMA standalone package [30], and we kept only the one-to-one orthology relationships as they have a higher probability of being *bona fide* orthologs, more likely to conserve their functions [42]. We kept 1117 one-to-one orthology relationships for *S. coelicolor* out of the total 2480 (45%), 661 out of 1480 (45%) for *B. subtilis*, and 641 out of 1488 (43%) for *E. coli* (Figure 4b). As expected, there was a greater number of one-to-one orthologous genes with *S. coelicolor* due to its phylogenetic closeness compared with the other two source organisms. Just by filtering orthologs, we restrained more than 50% of nodes to be included in the transferred interactions. The next filter is due to the completeness of the source networks since we can only transfer interactions between nodes already present in the source networks (Figure 4c). From there, we were primarily interested in TFs (white inner circles in Figure 4c), but we only considered those with at least one TG with a one-to-one ortholog in *C. glutamicum*, resulting in a total of 8, 7, and 13 potential TFs/regulons to be transferred from *S. coelicolor, B. subtilis*, and *E. coli*, respectively (colored inner circles in Figure 4c). However, the number of potential interactions to be transferred was reduced when we searched for a TF binding site in the promoter sequences of the orthologous TG in *C. glutamicum*; 24 out of the 479 interactions from the *S. coelicolor* network were conserved, along with the TF binding site, 17 out of 2576 from the *B. subtilis* network, and 70 out of 4653 from the *E. coli* network. We recovered more regulogs from *E. coli* due to the completeness of the source network. We lost many interactions through the stringent filters we applied, but we expect these conserved interactions to be true-positives. As mentioned above, the main goal of this interactions transfer is to detect interactions for the *C. glutamicum* TFs that are still missing in the network (Appendix A) despite the exhaustive work of the community to model the network. We retrieved interactions for a total of five DNA-binding TFs not considered in the current curation state of the network (Figure 4e). Given that the *C. glutamicum* regulatory mechanism is already one of the most studied and curated (Appendix A), most of the TFs that were retrieved from regulogs were already present in the *all evidence* network (Figure 4e). However, in terms of interactions, 82 out of the 111 interactions were not present in any of the *C. glutamicum* curated networks (Figure 4f). There was poor overlap between the regulogs obtained from each organism. There was one common TF between *E. coli* and *S. coelicolor* (Zur) and another one between *E. coli* and *B. subtilis* (LexA) (Figure 4g,h).

In the following section, we describe some of the conserved regulations in *C. glutamicum*. From *S. coelicolor*, 24 interactions were conserved. The interaction of Zur (*cg2502*) regulating *cg0042* is already part of the *strong* network (Appendix A). Another interaction is by RegX3 (*cg0484*), an essential response regulator of the SenX3–RegX3 two-component system [43]. RegX3 has a one-to-one orthology relationship with PhoP (*SCO4230*) from *S. coelicolor,* as does the gene *amtB* (*cg2261*) with *SCO5583*, for which a regulatory site for the PhoP ortholog in their upstream region is conserved. However, the interaction could not be transferred from *E. coli* or *B. subtilis* because a many-to-many orthology relationship was found for RegX3 in both organisms and, therefore, discarded. RegX3 has been characterized as a gene coding a regulator of phosphate-dependent gene expression in *Mycobacterium smegmatis* [44], required for virulence in *M. tuberculosis* [45], but its regulon has not been characterized in *C. glutamicum*. PhoP represses *amtB* and other nitrogen genes in *S. coelicolor* [46]. Previous work showed that *amtB* is required for ammonium uptake in *C. glutamicum* [47]. A binding site for PhoP was found 87–69 bp upstream of the *cg2261* translation start codon. This agrees with the mechanism of *amtB* regulation in *S. coelicolor*, binding upstream of the CDS and repressing its transcription by regulating a promoter in the upstream sequence from the binding site [46]. From *B. subtilis*, 17 interactions were fully conserved. For example, an autoregulation for LexA that was already part of the *strong* network (Appendix A). Cg1098 is an ortholog of SCO3129, a TetR family regulator involved in *S. coelicolor* osmotic stress [48]. In *S. coelicolor*, it regulates the transcription of two (*SCO3128* and *SCO3130*) genes and its own. However, only the autoregulation was fully conserved in *C. glutamicum*. Most of the characterized TetR family regulators regulate their own transcription [49].

From *E. coli*, we recovered a total of 70 interactions, for example, ArgR regulating *argC* (Appendix A), LexA (cg2114) regulating *recA* (Appendix A), and NrdR regulating *nrdI* (Appendix A). While the first two interactions are already included in the *strong* network, the latter is only included in the *all evidence* network. The gene *cg1327* has *b1334* as an ortholog, coding for the FNR global regulator in *E. coli*. For this protein, the regulation of *hmp* (*cg3141*) and the autoregulation were fully conserved. However, the *cg1327* gene is currently part of the basal machinery in the *C. glutamicum* network due to unreported characterization of its regulon. The gene *cg2899* codes for a regulator of the LysR family and is an ortholog of *b2537* (HcaR) in *E. coli*, regulating *hcaE*, which is an ortholog of *cg2637* (*benA*) in *C. glutamicum*, only regulated by GlxR and BenR in the *all evidence* network. In contrast with *C. glutamicum*, in *E. coli*, *hcaR* and *hcaE* are divergently transcribed, sharing the same promoter recognized by HcaR. The gene *cg0350* encodes for the GlxR ortholog to CRP in *E. coli*, both being global regulators in their corresponding networks. The regulation of CRP to *dadA* (*b1189*) is fully conserved in *C. glutamicum* for their orthologs (GlxR and its target *cg3340*, repectively). The gene *cg3340* is currently regulated only by SigA. The other TG conserved is *cg2175* (with *b3167* as its ortholog in *E. coli*), which codes for a ribosome binding protein. However, none of the two targets were identified in a previous *in silico* analysis of the GlxR regulon in *C. glutamicum* [50]. The gene *cg1425*, coding for LysG (ArgP encoded by *b2916* in *E. coli*), regulates *dnaA* that is not part of the current *C. glutamicum* network. However, none of the three interactions were conserved in *C. glutamicum*. DnaA, besides being the protein for DNA replication initiation, is a transcriptional regulator that controls the transcription of its own coding gene and at least 10 others in *E. coli*. The autoregulation and the regulation of the other four genes (*cg0004*, *cg0005*, *cg1525*, and *cg1550*) were fully conserved in *C. glutamicum* (Appendix A). Zur is encoded by *cg2502*, ortholog to *b0683* in *E. coli*. A regulation from Zur to *cg2183* was recovered from the *oppC* gene in *E. coli*. The interactions are not part of the current networks for *C. glutamicum*. LldR is encoded by *cg3224*, ortholog to *b2980* (*glcC* in *E. coli*), which regulates *glcB*. The interaction was conserved in *C. glutamicum* but not present in the current networks, although the LldR regulon has 12 TGs already.

These results show that even though some interactions that are already known in *C. glutamicum* are recovered, the rate of recovered interactions is low. Therefore, for long phylogenetic distances, it might be better to discriminate false-positives after a mildly lax prediction. We noticed that most of the interactions are lost due to the conservative approach of using only one-to-one orthologs. A potential solution for this is the use of other orthology relationships, with subsequent discrimination of false-positives through the conservation of regulogs not only in *C. glutamicum* but also in other closely related organisms, conferring greater confidence values to those interactions highly conserved.

## 4. Conclusions

In this work, we update the *C. glutamicum* regulatory network by manual curation of the literature. We also went beyond the regulation of transcription initiation to incorporate regulations mediated by protein–protein interactions and small RNAs. Three network models with different confidence levels were reconstructed and deposited in the new v2.4 of Abasy Atlas (https://abasy.ccg.unam.mx (accessed on 1 January 2021)). Poor efforts have been carried out to provide consolidated, disambiguated, homogenized high-quality regulatory networks on a global scale, with their structural properties, system-level components, and historical snapshots to trace their curation process. We originally conceived Abasy Atlas to fill this gap by making a cartography of the functional architectures of regulatory networks for a wide range of bacteria.

This work provides the most complete and reliable set of *C. glutamicum* regulatory networks, which can be used as the gold standard for benchmarking purposes and training data for modeling. The *C. glutamicum* regulatory networks have been metacurated to avoid heterogeneity such as inconsistencies in gene symbols and heteromeric regulatory complexes representation. This enables large-scale comparative systems biology studies to understand the common principles and particular lifestyle adaptations of regulatory systems across bacteria and to implement those principles into future work such as the reverse engineering of regulatory networks. The historical snapshots deposited in Abasy Atlas allow us to carry out network analyses at different incompleteness levels, making it possible to identify how a methodology is affected, to pinpoint potential bias and improvements, and to predict future results. Regulatory network models, gene information, and module annotations can be downloaded from the “Downloads” section in Abasy Atlas (https://abasy.ccg.unam.mx/downloads (accessed on 1 January 2021)). The same web page provides useful information about the downloadable files.

## Figures and Tables

**Figure 1 microorganisms-09-01395-f001:**
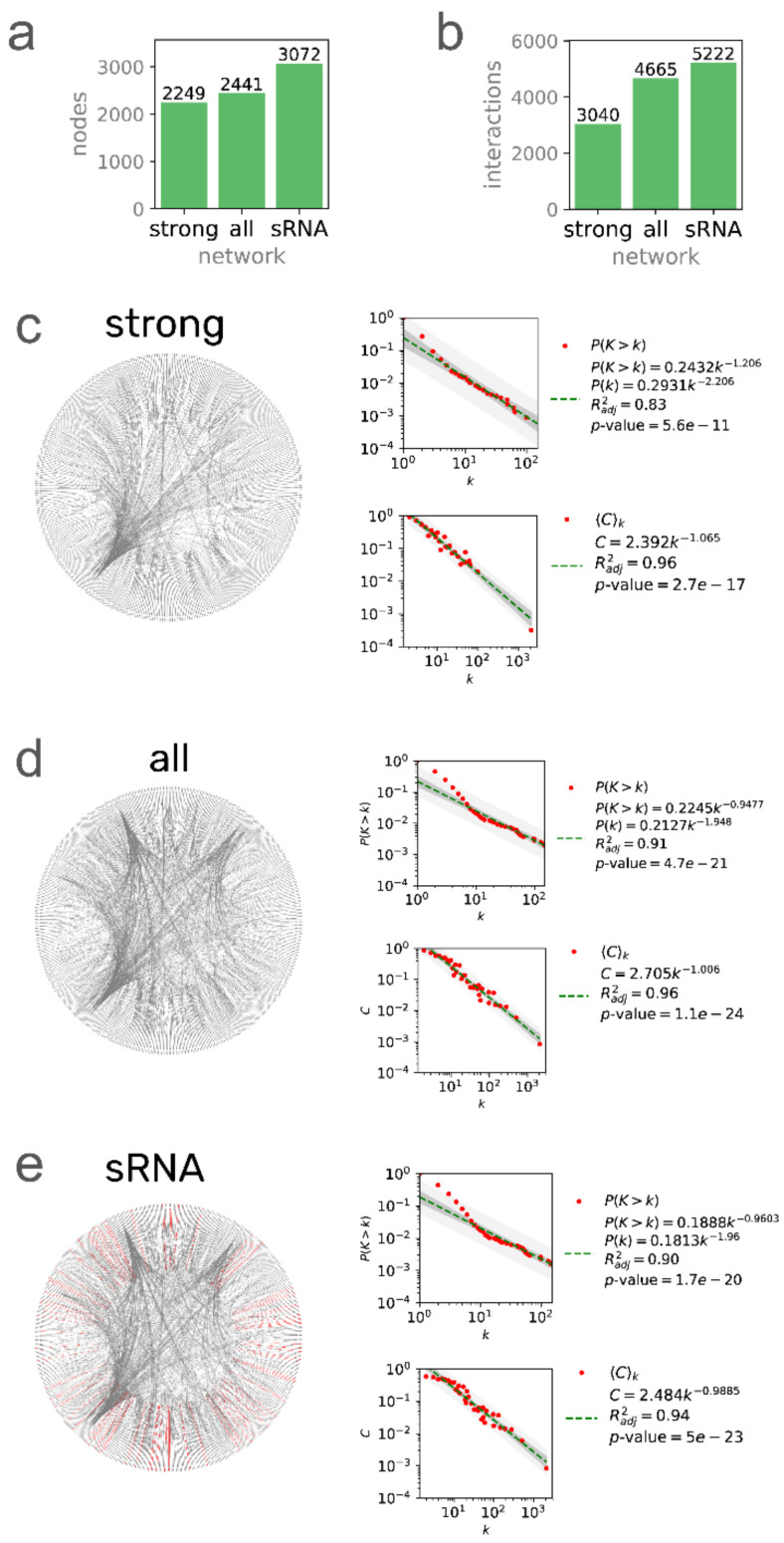
Three network models of the *C. glutamicum* regulatory network. The number of nodes (**a**) and interactions (**b**) for the three networks. Network, P(k), and C(k) distributions for (**c**) 196627_v2020_s21_eStrong (*strong*), (**d**) 196627_v2020_s21 (*all evidence*), and (**e**) 196627_v2020_s21_dsRNA (*sRNA*) networks. Network plots were generated with Circos [36] using the leftmost gene/sRNA coordinates to sort the nodes clockwise. Nodes with no coordinates in the genome annotation were disregarded.

**Figure 2 microorganisms-09-01395-f002:**
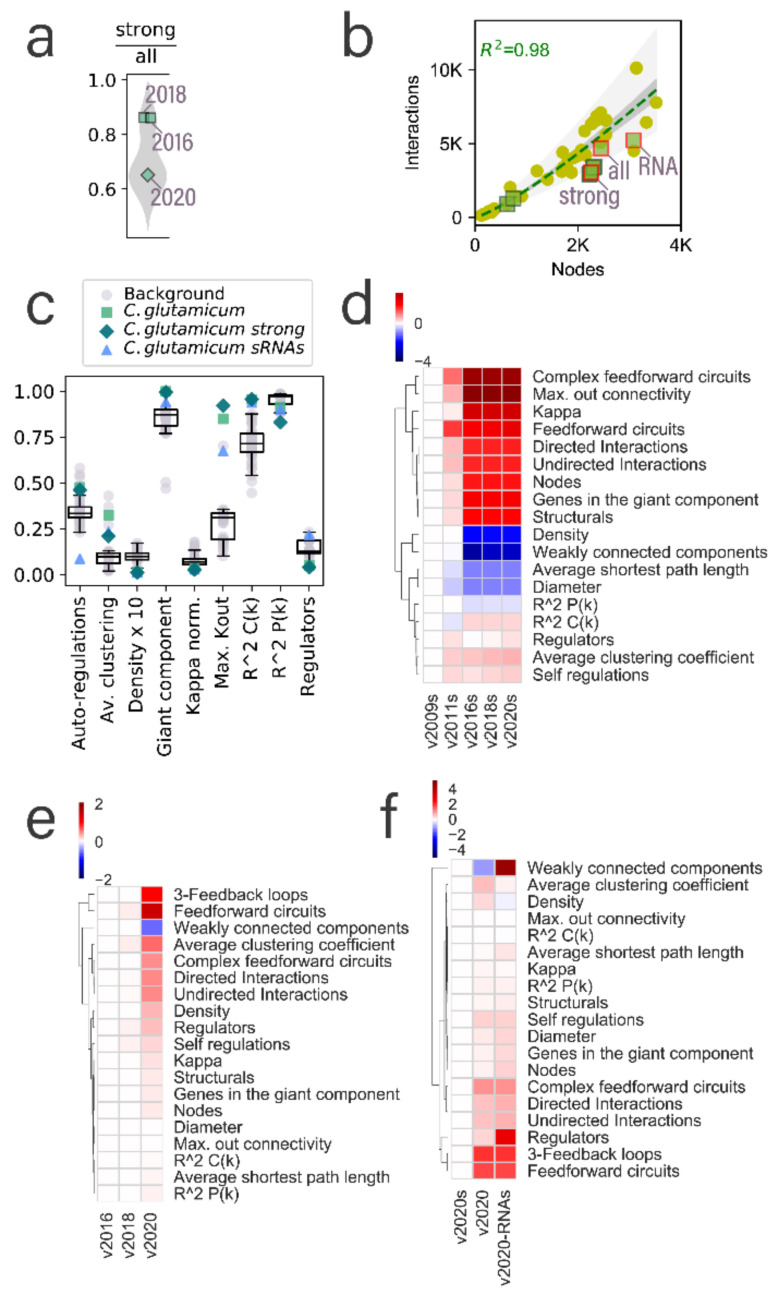
Structural properties of the *C. glutamicum* networks. (**a**) Distribution of the fraction of the strong interactions in the *all evidence* networks, including at least one strong interaction. *C glutamicum* networks are highlighted and labeled. 2009 and 2011 versions of the *C. glutamicum* network are not included as they do not have a cognate *all evidence* network. (**b**) Inclusion of the three networks presented in this work into the previous model reported in [1] for the inference of the number of interactions for the regulatory networks. *C. glutamicum* networks are marked with green squares, and the three networks reported in this work are highlighted with a red outline and labeled. The rest of the data points (yellow dots) are the rest of the Abasy Atlas database used for reference. (**c**) Comparison of *C. glutamicum* structural properties with the nonredundant set of bacterial networks used as background. Boxplots were drawn, including the nonredundant data set and the *C. glutamicum* networks reported in this work. (**d**) Heatmap values are the log2-fold change of the *C. glutamicum* regulatory networks for *strong* networks of versions 2011, 2016, 2018, and 2020, relative to the earliest *strong* version (2009). The v2009 column is included for clarity. Properties are clustered to ease the identification of those that have increased, decreased, or remained virtually unchanged. Heatmaps (**e**,**f**) also represent the log2-fold change values relative to the leftmost column of (**e**) for versions of the *all evidence* network and (**f**) for the three different network models presented in this work to highlight the impact of the inclusion of sRNA-mediated interactions into the structural properties of the network.

**Figure 3 microorganisms-09-01395-f003:**
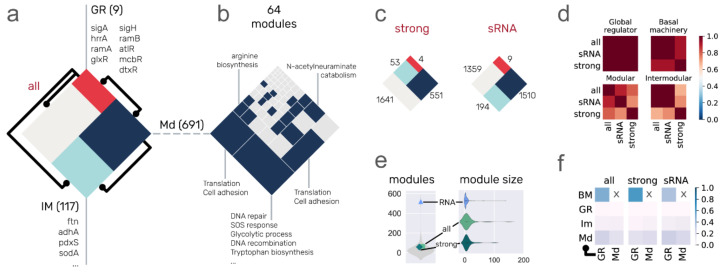
System-level classification of the networks. (**a**) The diamond represents the complete set of nodes in the network, which are classified in one of the four classes: global regulators (red), modular (dark blue), intermodular (light blue), and basal machinery (gray, 1624 nodes). The size of the classes is proportional to the size of the *all evidence* network on a logarithmic scale. Black lines represent the interactions between the two classes. We listed the global regulators and some examples of intermodular genes. The modular class is further divided into 64 locally independent modules in the *all evidence* network (**b**). Modules enriched with a biological function are colored in blue. The size of the sections is proportional to the size of the modules. Similar to the *all evidence* network in panel (**a**), panel (**c**) shows the proportion of the NDA classes for the *strong* and *sRNA* networks. (**d**) Heatmaps of similarity index between the three *C. glutamicum* networks for each one of the four NDA classes. The color bar shows that more than half of the nodes in the class are conserved for each class among the three networks, showing the precision of network node classification. (**e**) Distribution of the number of modules and their size. Light gray distribution of the number of modules was drawn using the nonredundant set of networks, including the three *C. glutamicum* networks. (**f**) The fraction of network interactions between the four classes for each one of the *C. glutamicum* networks.

**Figure 4 microorganisms-09-01395-f004:**
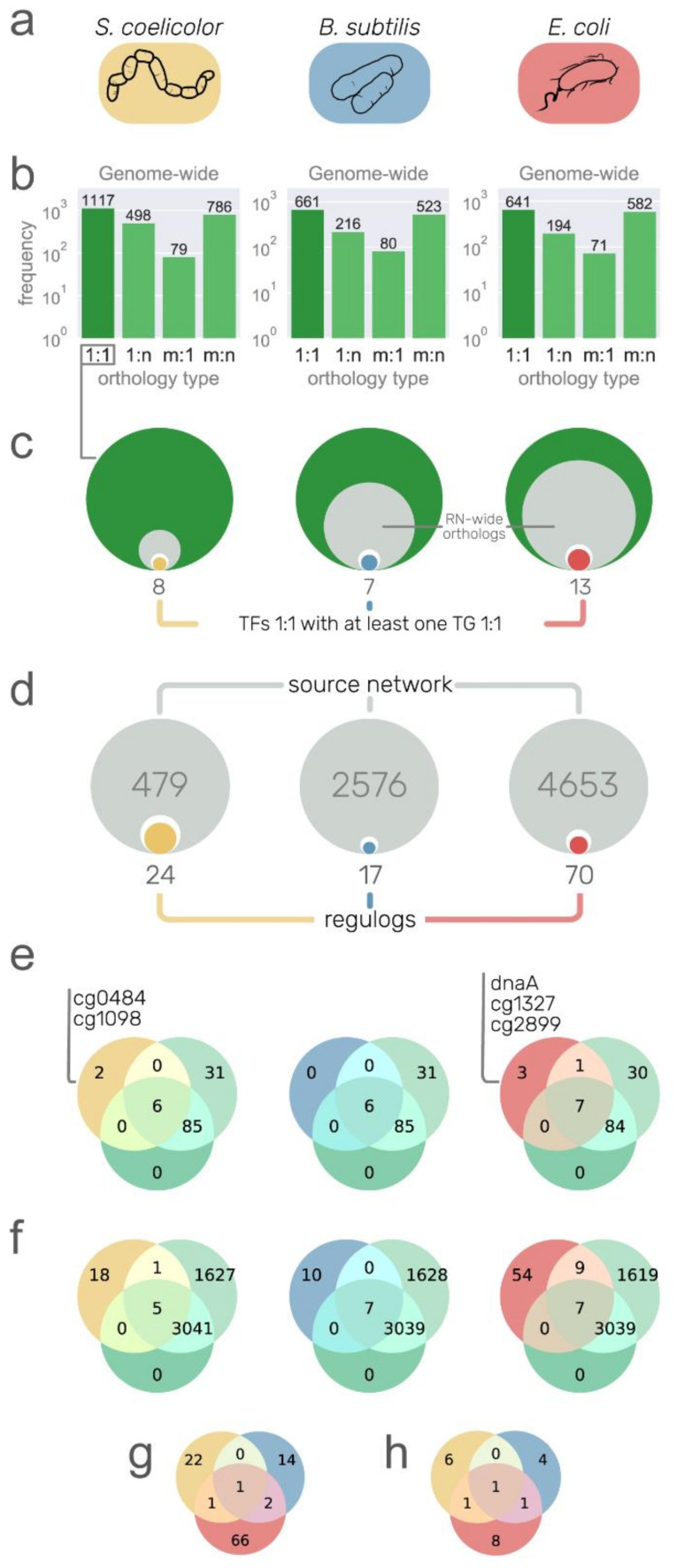
Putative regulons from other model organisms. (**a**) Networks with strong interactions of *S. coelicolor, B. subtilis*, and *E. coli*. used as a source of information. Rounded rectangles color is used to relate the organism to the rest of the figure. (**b**) Orthology relationship type between source organisms and *C. glutamicum*. Only one-to-one relationships were used for downstream analysis. (**c**) Size comparison between the one-to-one orthology genes (green circles), the orthologs with at least one interaction in the source network (inner gray circle), transcription factor (TF) orthologs (inner white circle), and TF orthologs with at least one target gene (TG) with one-to-one orthology relationship (inner colored circles and numbers). (**d**) Size comparison between the source networks (gray circles with large gray numbers), TF–TG pairs conserved as orthologs one-to-one in *C. glutamicum* (inner white circles), and the interactions conserved with a TF binding site in the promoter region of the TG (colored inner circles and numbers of regulogs). (**e**) Venn diagrams showing the overlap of TFs between three sets: the *strong* network (green circle), the *all evidence* network (light green circle), and the interactions from the source organisms with the unique TFs listed. (**f**) Venn diagrams showing the overlap of interactions between the *strong* network, the *all evidence* network, and the regulogs network from source organisms. (**g**,**h**) Euler diagrams showing poor overlap between the regulogs (**g**), and their TFs (**h**).

## Data Availability

The data sets supporting the results of this work are publicly accessible from Abasy Atlas (https://abasy.ccg.unam.mx (accessed on 1 January 2021)).

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
