# Peer review of "Corynebacterium glutamicum Regulation beyond Transcription: Organizing Principles and Reconstruction of an Extended Regulatory Network Incorporating Regulations Mediated by Small RNA and Protein–Protein Interactions"

_microorganisms, 2021, doi:10.3390/microorganisms9071395_

Round 1

Reviewer 1 Report

The manuscript describes construction of three types of Corynebacterium glutamicum regulatory networks based on the data available in the literature. Further, the network is compared with the known networks of other bacteria.

Major comments:

1) The quality of English should be considerably improved. There are many clumsy expressions and unclear sentences missing sometimes logic, particularly in Abstract and Introduction. I recommend the authors to use professional English-language editing and proofreading service.

2) Introduction mainly provides description of properties of the constructed regulatory networks and procedure how to construct them. It is not clear why the section 1.1. “Analyzing regulatory networks: a primer” is separated and what exactly is meant by the “primer”. I would expect that the introduction would describe what practical or theoretical problems can be solved using such constructed networks and what particular question has already been solved using the previous Corynebacterium glutamicum regulatory networks. In other words: How can these models of regulatory networks be used by the researchers, who are not specialists in this area.

3) The research field is very specific and it is not easy for the non-experts to understand the procedure and the practical output of the results. The authors should make this easier. They should have in mind that the take-home-message must be given in much more comprehensible form. Although it is not a usual point in research papers, I recommend to add a dictionary of the terms used in the study and their clear definitions.

4) The authors should provide in the end short instructions how to use the models and what kind of problems and questions may be solved using them.

Minor comments:

Lines 15-17” Too long sentence, please rephrase

Line 19: What is meant by "any"?

Line 20: “regarding” – compared to?

Line 21: It is not clear who did the “directed experiments”.

Line 21: It is not apparent what is “second model”, “third model” (previous?), what are the other models and how they differ

Line 23-25: Please, rephrase the sentence “We study…properties”

Line 25: The “system-level components” should be defined

Line 28: What is “strong network”?

Line 34: “bacterium with a complex regulatory network” this expression provides no information. Every organism has complex regulatory network

Line 36: Line 36: “model” for what?

Line 39: However, like S. coelicolor…. also tethers oriCs to the cell poles.” Why this special information is provided?

Line 48: “..scenario”… for what?

Line 47: “..networks still..”  “are” is probably missing

Line 19: “…networks, it results in biased structural properties..” – please, rephrase

Line 53: “Prove of that” – probably “Proof”

These are only examples of unclear expressions and such comments could continue at least till line 200.

Line 203: Please, explain first what three models were constructed and what is the basic difference between them.

Line 258: upstream of what?

Line 270: Proteins coded by…genes were classified …

Line 270-271: “basal machinery, modular, intermodular” Are these classes of proteins? Please, reword.

Line 274-275: “.. value was…and removed”. It would be better to use past tense at all places

Line 289: “…goodness…”.. reword

Line 300: What is “giant component?”

Author Response

The manuscript describes construction of three types of Corynebacterium glutamicum regulatory networks based on the data available in the literature. Further, the network is compared with the known networks of other bacteria.

Major comments:

1) The quality of English should be considerably improved. There are many clumsy expressions and unclear sentences missing sometimes logic, particularly in Abstract and Introduction. I recommend the authors to use professional English-language editing and proofreading service.

R. Thank you. We have improved the writing of the whole manuscript.

2) Introduction mainly provides description of properties of the constructed regulatory networks and procedure how to construct them. It is not clear why the section 1.1. “Analyzing regulatory networks: a primer” is separated and what exactly is meant by the “primer”. I would expect that the introduction would describe what practical or theoretical problems can be solved using such constructed networks and what particular question has already been solved using the previous Corynebacterium glutamicum regulatory networks. In other words: How can these models of regulatory networks be used by the researchers, who are not specialists in this area.

R. We consider that the mentioned section requires to be delimitated from the rest as it provides the basics required to analyze regulatory networks at global scale. We changed the title of the section from “Analyzing regulatory networks: a primer” to “A primer on analyzing regulatory networks” to clarify. See also the answer to issue 4 below.

3) The research field is very specific and it is not easy for the non-experts to understand the procedure and the practical output of the results. The authors should make this easier. They should have in mind that the take-home-message must be given in much more comprehensible form. Although it is not a usual point in research papers, I recommend to add a dictionary of the terms used in the study and their clear definitions.

R. This is exactly the reason why we have included the section “A primer on analyzing regulatory networks”. We cited several papers that summarize the state of the knowledge and terms used in networks biology and boldfaced each term in our text to ease identification.

4) The authors should provide in the end short instructions how to use the models and what kind of problems and questions may be solved using them.

R. Applications for the three network models are described in section 3.1. “The regulatory networks of C. glutamicum and potential applications”. We included “and potential applications” to point it out. A few other applications are named in the second paragraph of the section “Conclusions”, and we appended a sentence to the same paragraph to mention how to retrieve the networks and useful related information such as gene id/symbol mapping and modules annotation.

Minor comments:

Lines 15-17” Too long sentence, please rephrase

R. Done.

Line 19: What is meant by "any"?

R. At least one experimental evidence. We used “any” to avoid innecesary wording.

Line 20: “regarding” – compared to?

R. Changed as suggested.

Line 21: It is not clear who did the “directed experiments”.

R. We now specify that the interactions have been previously identified.”… regulations previously supported by directed experiments…”.

Line 21: It is not apparent what is “second model”, “third model” (previous?), what are the other models and how they differ

R. We rephrased this section of the abstract to clarify the types of interactions considered in each of the tree network models and how they differ from each other. We also using absolute values reported in the abstract and changed the value for the interactions included in this version to the previous one (1225 instead of 557 which is the number of interactions in the sRNA network not part of the all evidence network). We apologize for the misleading interpretations the previous form may have caused.

Line 23-25: Please, rephrase the sentence “We study…properties”

R. Done.

Line 25: The “system-level components” should be defined

R. Due to the word limit for the abstract (up to 200) we only define it in the main text.

Line 28: What is “strong network”?

R. Those networks that were constructed solely with strongly-supported interactions (validated by directed experiments). Now we define the strong network of C. glutamicum before in the abstract to be used as a reference for the other organisms (on the cited line).

Line 34: “bacterium with a complex regulatory network” this expression provides no information. Every organism has complex regulatory network

R. We rephrased.

Line 36: Line 36: “model” for what?

R. for the study of regulatory networks. We rephrased: “…It is also a model organism for the study of regulatory networks..”.

Line 39: However, like S. coelicolor…. also tethers oriCs to the cell poles.” Why this special information is provided?

R. We removed the sentence.

Line 48: “..scenario”… for what?

R. We rephrased: “The network incompleteness situation is worst for non-model organisms for which little or none is known about their transcriptional machinery”.

Line 47: “..networks still..”  “are” is probably missing

R. Right, it was missing. Thank you.

Line 19: “…networks, it results in biased structural properties..” – please, rephrase

R. Done.

Line 53: “Prove of that” – probably “Proof”

R. Changed as suggested.

These are only examples of unclear expressions and such comments could continue at least till line 200.

We have proof-read the manuscript.

Line 203: Please, explain first what three models were constructed and what is the basic difference between them.

R. We moved the sentence “The three networks reconstructed in this work were deposited in the new v2.4 of Abasy Atlas.” to the end of the same paragraph, after the definition of the networks and their differences in terms of the type of interactions considered for their reconstruction.

Line 258: upstream of what?

R. With reference to the start codon. We rephrased the sentence as follows: “Upstream (up to -300 to +50 ) sequences with reference to the start codon for the four genomes were retrieved from the RSAT suite [21] with the retrieve-seq tool preventing overlap with neighboring genes.”

Line 270: Proteins coded by…genes were classified …

R. Thanks for pointing it out. We changed it to “Nodes were classified into one of the four classes” as nodes may be genes, proteins, or sRNAs.

Line 270-271: “basal machinery, modular, intermodular” Are these classes of proteins? Please, reword.

R. Classes of nodes in the network (either genes, proteins, or sRNAs). We rephrased the sentence.

Line 274-275: “.. value was…and removed”. It would be better to use past tense at all places

R. Changed as suggested.

Line 289: “…goodness…”.. reword

R. We rephrased the sentence to improve clarity. Although we did not change the phrase “goodness of fit” as it is the standard term to describe how well a model fits the observations.

Line 300: What is “giant component?”

R. We changed the sentence to “The size of the giant component was normalized by…” as the value (size) is the one being normalized, not the giant component itself. We included the definition “The giant component is the largest component of the network, and its size is determined by the number of nodes it covers.” in the background.

Reviewer 2 Report

The authors herein provided a detailed improvement in the field of system biology of C. glutamicum. I would thank the editor and authors to allow me to read the manuscript. The paper is well written and no specific requirements are needed.

Author Response

The authors herein provided a detailed improvement in the field of system biology of C. glutamicum. I would thank the editor and authors to allow me to read the manuscript. The paper is well written and no specific requirements are needed.

R. We thank the anonymous reviewer for his/her assessment of our work.

Round 2

Reviewer 1 Report

The manuscript was improved considerably.